# Smartphone-Guided Educational Counseling and Self-Help for Chronic Tinnitus

**DOI:** 10.3390/jcm11071825

**Published:** 2022-03-25

**Authors:** Winfried Schlee, Patrick Neff, Jorge Simoes, Berthold Langguth, Stefan Schoisswohl, Heidi Steinberger, Marie Norman, Myra Spiliopoulou, Johannes Schobel, Ronny Hannemann, Rüdiger Pryss

**Affiliations:** 1Department of Psychiatry and Psychotherapy, University of Regensburg, 93053 Regensburg, Germany; patrick.neff@uzh.ch (P.N.); jorge.simoes@ukr.de (J.S.); berthold.langguth@medbo.de (B.L.); stefan.schoisswohl@ukr.de (S.S.); heidi.steinberger@stud.uni-regensburg.de (H.S.); m.norman@gmx.at (M.N.); 2URPP (University Research Priority Program) “Dynamics of Healthy Ageing”, University of Zürich, 8006 Zürich, Switzerland; 3Knowledge Management and Discovery Lab, Otto-von-Guericke University, 39106 Magdeburg, Germany; myra@ovgu.de; 4DigiHealth Institute, Neu-Ulm University of Applied Sciences, 89231 Neu-Ulm, Germany; johannes.schobel@hnu.de; 5WSAudiology, Sivantos GmbH, 91058 Erlangen, Germany; ronny.hannemann@wsa.com; 6Institute of Clinical Epidemiology and Biometry, University of Würzburg, 97080 Würzburg, Germany; ruediger.pryss@uni-wuerzburg.de

**Keywords:** tinnitus, self-help, ecological momentary assessment, ehealth, smart-phone, intervention

## Abstract

Tinnitus is an auditory phantom perception in the ears or head in the absence of a corresponding external stimulus. There is currently no effective treatment available that reliably reduces tinnitus. Educational counseling is a treatment approach that aims to educate patients and inform them about possible coping strategies. For this feasibility study, we implemented educational material and self-help advice in a smartphone app. Participants used the educational smartphone app unsupervised during their daily routine over a period of four months. Comparing the tinnitus outcome measures before and after smartphone-guided treatment, we measured changes in tinnitus-related distress, but not in tinnitus loudness. Improvements on the Tinnitus Severity numeric rating scale reached an effect size of 0.408, while the improvements on the Tinnitus Handicap Inventory (THI) were much smaller with an effect size of 0.168. An analysis of user behavior showed that frequent and intensive use of the app is a crucial factor for treatment success: participants that used the app more often and interacted with the app intensively reported a stronger improvement in the tinnitus. Between study allocation and final assessment, 26 of 52 participants dropped out of the study. Reasons for the dropouts and lessons for future studies are discussed in this paper.

## 1. Introduction

Tinnitus is an auditory phantom perception in the ears or head in the absence of a corresponding external stimulus [1,2]. With a prevalence of 8–28%, the condition is very common in Western societies [3,4], and 1–4% of people report that they severely suffer from tinnitus [3,5,6]. In those severe cases, tinnnitus is frequently associated with depression, anxiety, insomnia, concentration difficulties, and poor psychological health, all of which have a significant impact on their quality of life [5,6]. The term “tinnitus disorder” has recently been suggested to describe these severe cases where auditory phantom perception is associated with subjective suffering of the affected people [2].

Currently, there is no effective treatment available that reliably reduces auditory perception of tinnitus. Reviewing the progress of clinical research in the field of tinnitus, several authors [7,8,9,10] highlighted the large patient heterogeneity as one of the leading reasons for inconsistent results in clinical research hampering the development of a uniformly effective treatment for tinnitus. In a review by Baguley et al., at least 13 different types of causal factors for tinnitus were identified [6]. It is supposed that these different etiologic factors result in various subtypes of tinnitus with differences in their pathophysiology, which in turn may require different treatment strategies. Even though effective treatments for reducing the sound perception need further research and development, effective treatments for reducing the subjective suffering already exist. In a recent Cochrane review, Fuller and colleagues [11] analyzed the effectiveness of Cognitive Behavioral Treatment (CBT) and reported an average effect size of 0.56 for reducing the impact of tinnitus on quality of life at the end of treatment. CBT is an umbrella term that includes numerous psychological interventions that were developed to reduce the impact of tinnitus on the quality of life of the affected person. Cognitive strategies aim to replace irrational thinking styles by alternative thinking that leads to more adaptive responses. Behavioral therapies aim to overrule and modify maladaptive behavior.

Educational counseling is a treatment approach that aims to educate people with chronic tinnitus and inform them about possible coping strategies. By means of education, counseling, and informing the patient about tinnitus, the patient should be helped to better understand the underlying causes of the conscious tinnitus perception and to deal with possible accompanying symptoms such as sleep disturbances, concentration problems, etc. Such an educational counseling approach aims to promote self-help capacities of the patients and enhance patient empowerment. In principle, educational counseling can be applied as a solitary therapy or in combination with other clinical interventions [12].

In a multicenter randomized clinical trial, Henry and colleagues [13] enrolled 148 veterans with chronic tinnitus and randomized them for a tinnitus masking treatment (*n* = 42), tinnitus retraining therapy (*n* = 34), tinnitus educational counseling (*n* = 39), or a waiting list control (*n* = 33). Tinnitus severity was significantly reduced in the tinnitus masking, tinnitus retraining therapy, and tinnitus educational counseling group compared to the waiting list group directly after the end of treatment as well as after a six-month follow-up period. A comparison between the treatment groups did not reveal a difference of statistical significance, showing that educational counseling alone is similarly effective to tinnitus masking and the tinnitus retraining treatment. In a systematic review, Xiang et al. [14] analyzed nine clinical studies (the Henry study included) to compare stand-alone educational counseling with other psychological interventions or combination therapies and did not find a significant difference between the interventions. Even though the clinical practice guidelines (CPG) of the American Academy of Otolaryngology-Head&Neck Surgery foundation [15] as well as the European Tinnitus Guidelines [16] recommend educational counseling as one of the treatment options for chronic tinnitus, systematic research on educational counseling in tinnitus is still limited.

The aim was to test the feasibility of app-based educational counseling treatment on a clinical tinnitus population where the participants use the smartphone unsupervised during their daily routine. In this study, we implemented educational material and self-help advice in a smartphone app and measured tinnitus symptoms daily for a period of four months. Every day, the participants received new advice for tinnitus self-management together with a medical explanation of the advice. With this study, we aimed to investigate the clinical improvement induced by an electronic tool for systematic educational counseling and self-help in tinnitus and to research the relationship between active user involvement and clinical improvement. 

There are three major challenges with studies on educational counseling. The first challenge is the choice of a control condition. Since the concept of the TinnitusTipps App involved daily tips, the participants were confronted with their tinnitus on a daily basis and were motivated to actively think about their tinnitus, their coping strategies and how to improve them. A study design with the aim to test the impact of the tips, should contain a control group that is also confronted with the tinnitus on daily basis and actively think about their tinnitus—but without receiving the specific tinnitus tips of the app. In the first run of this study, we implemented a control group that also received daily questionnaires that ask the participant to reflect about the individual tinnitus perception. However, no tips for improving the coping strategies were given. The second challenge is the standardization of the intervention across participants. In the clinical setting, counseling is typically provided in social situations where a therapist provides the educational material. Depending on the quality and the duration of this social interaction, some participants receive more and some participants less information. In the presented study, we standardized the counseling with the smartphone app “TinnitusTipps” that provided only one tip per day in the same way for all participants. The third challenge is the assessment of the active cognitive involvement of the individual participants in the educational process. How can we know that the participant concentrates on the counseling session and actively thinks about the content? Especially with an unsupervised counseling app like the Tinnitus Tipps app, the active engagement of the participants might largely vary. In this study, we assessed the amount of app use by the total number of session and the active engagement by the number of tip ratings provided by the user.

## 2. Materials and Methods

### 2.1. Study Design and Participants

For this feasibility study, we report two samples that have used the same app-based educational counseling with similar study design outcome assessment (Figure 1). In the first run, participants were randomized to treatment arms 1 and 2. In treatment arm 1, the users received four months of smartphone-guided educational counseling treatment plus Ecological Momentary Assessment (EMA). In treatment arm 2, participants received for two months EMA only, followed by a period of another two months where the users received the smartphone-guided educational counseling treatment plus EMA. In the second run (treatment arm 3), there was no randomization and all participants received four months of smartphone-guided educational counseling treatment plus Ecological Momentary Assessment (EMA). The study protocol was reviewed by the Research Ethics Committees of the University Hospital Regensburg (protocol number: 17-544_1-101). The app was implemented on iOS devices.

Participants were contacted by email or telephone and invited to participate in this study. Participants were selected from a list of persons with chronic tinnitus from the tinnitus center in Regensburg, which had previously indicated their willingness to participate in studies on tinnitus. Only adults with chronic tinnitus for at least 6 months were included in the study. The participants needed a smartphone running an iOS system greater than 10. Candidates were excluded in the case of an acute psychosis, an acute major depression or a substance abuse disorder in the 12 weeks before starting treatment, epilepsy or other diseases of the central nervous system, or if they had undergone another treatment within three months before treatment start. Participants with pharmacological treatment were only included if the medication was stable at least 10 days before treatment start. 

### 2.2. Smartphone-Guided Educational Counseling

During the treatment phase, participants received every day a “tip of the day”. The tips were randomly assigned from a library of 108 self-help tips and the tip was displayed after completion of the EMA questionnaire. Two examples of the daily tips are given in the Appendix A.

The tips were designed to be concise and succinct so that they would preferably not exceed the display size of an average smartphone and that reading the tips would be possible without spending a lot of time in everyday life. Furthermore, the tips should also be as detailed and informative as possible in order to convey the content to the user in an understandable way. For this reason, the tips were structured in a uniform manner: Each self-help tip consists of a title, an objective, the tip itself, and a more detailed explanation of the tip. The “title” helps the respondent find a tip again if needed and gives a brief insight into the content of the tip. The “objective” summarizes the tip in one sentence. The section with the actual tip contains the essential information, recommendations, and hints. In the explanation section, the reader can find background knowledge or supplementary information about the tip. This structure allows the respondent to quickly decide which sections of the tip are relevant and worth reading, depending on the available time and interest. The tips were on average 101 words long (range 40 to 155 words). The tips covered topics such as, e.g., the influence of sport on tinnitus, self-help groups, the use of noizers, cognitive appraisal, influence of emotions and how to modulate, music listening, relaxation of jaw and neck or noise protection. The tips were developed based on existing self-help guides for tinnitus [17,18,19,20,21] and underwent an internal review process before being implemented in the app. 

The smatphone app was used in the testflight mode of iOS, which is a testing environment for smartphone apps. The apps in the testflight do not appear in the official app store and have to be installed in a different way. To install the app, participants received an email with a link to install the “TestFlight” app und iOS. The “TestFlight” app in turn enabled the installation of the TinnitusTipps app. When opening the TinnitusTipps app for the first time, the user had to log in with their account credentials. Then they received the invitation to the respective study group, which had to be accepted. As support, study participants were sent an email explaining the installation process both in written form and in the form of a short video.

### 2.3. Ecological Momentary Assessment

Ecological Momentary Assessment (EMA) of the tinnitus was made using two questions on tinnitus loudness and tinnitus distress, using the questionnaire published by Schlee et al., 2016 [22]. In addition, participants were asked about their momentary hearing ability, their limitation due to hearing loss, and if they were using a hearing aid at the moment. Furthermore, they were asked about their current perceived stress level, and exhaustion. The goal was to fill out the EMA questionnaire three times per day and the participants received random notifications to fill out the questions. The EMA data are reported in [23] and a screenshot of the daily questionnaire is provided in the Appendix A.

### 2.4. Measurement Instruments

The Tinnitus Handicap Inventory (THI) [24] and the Tinnitus Severity Numeric Rating Scale [7] are clinical outcome measures that are commonly used to assess tinnitus distress. The second question of the Tinnitus Severity Numeric Rating Scale is used to collect self-ratings of the subjective tinnitus loudness. The THI questionnaire served as the primary outcome measure for this study. All tinnitus questionnaires were assessed before the beginning of the study and after the end of the four month study.

The European School for Interdisciplinary Tinnitus Research Screening Questionnaire (ESIT-SQ) is an instrument to systematically assess the tinnitus medical history [25]. At the time of data collection, no validated tool was available to consistently measure a patient’s level of empowerment. Thus, for the present study, the Tinnitus Empowerment Scale (TES) was designed based on patient empowerment questionnaires for diabetes [26] and psoriasis [27]. In this questionnaire, three questions were asked about tinnitus literacy to assess how much the patient knows about tinnitus; three questions were asked about coping skills for tinnitus management; three questions assessed how self-confident the user is when thinking and talking about tinnitus; three questions were asked about self-confidence in coping with the tinnitus; and three questions were asked about the awareness of the cause for tinnitus and factors that influence the dynamic changes in one’s own tinnitus. An English translation of the questionnaire is provided in the Appendix A. 

### 2.5. Statistical Analysis

Statistical analyses were conducted in R [28], using R-version 4.0.3 (10 October 2020). Mixed model analysis was calculated using the nlme package (version 3.1-152). Handling of the data from the smartphone app were done using the jsonlite package (version 1.7.2). Missing values were coded as NA and removed for the respective analysis. The non-parametric Kruskal-Wallis test was used for analyzing continuous variables whether the participants in the three arms originate from the same patient distribution. Chi square tests were used to analyze count variables. Cohen’s d effect size was calculated using the effsize package.

## 3. Results

### 3.1. Characteristics and Summary of Study Participants

In total, 36 people with chronic tinnitus (14 female, 22 male) with an average age of 49.4 (SD 11.7) years finished the smartphone-guided educational treatment and were included in the final analysis. More details about the study groups at baseline are given in Table 1.

### 3.2. Within-Arm and between-Arm Analysis of Tinnitus Symptoms

THI sum score. A two-way mixed model ANOVA was performed to analyze the effect of the time point (pre vs. post) and group allocation on the THI sum score (Table 2). Analysis of the main effects revealed that group allocation did not have a statistically significant effect on the THI sum score (F(2, 33) = 0.270, *p* = 0.765) while the influence of the time point was close to the threshold of statistical significance (F(1, 31) = 3.94, *p* = 0.056). The analysis revealed that there was no statistically significant interaction between the time point and group allocation (F(2, 31) = 0.403, *p* = 0.672).

For post hoc analysis, a Wilcoxon Signed Rank Test was calculated and revealed statistical significance between the pre- and post-intervention phase (W = 345, *p* = 0.02). Cohen’s d effect size was calculated with 0.168 (confidence interval: −0.31 to 0.64). 

Tinnitus severity 1: problem of tinnitus. A two-way mixed model ANOVA was performed to analyze the effect of the time point (pre vs. post) and group allocation on the TS1 score (Table 2). Analysis of the main effects revealed that group allocation did not have a statistically significant effect on TS1 score (F(2, 33) = 0.862, *p* = 0.432) while the influence of the time point was statistically significant (F(1, 31) = 14.0, *p* = < 0.001). The analysis revealed that there was no statistically significant interaction between the time point and group allocation (F(2, 31) = 3.14, *p* = 0.057).

For post hoc analysis, a Wilcoxon Signed Rank Test was calculated and revealed statistical significance between the pre- and post-intervention phase (W = 119, *p* = 0.003). Cohen’s d effect size was calculated with 0.408 (confidence interval: −0.07 to 0.89). 

Tinnitus severity 2: tinnitus loudness. A two-way mixed model ANOVA was performed to analyze the effect of the time point (pre vs. post) and group allocation on the TS2 score (Table 2). Analysis of the main effects revealed that the group allocation did not have a statistically significant effect on the TS2 score (F(2, 33) = 0.265, *p* = 0.769) neither the time point (F(1, 31) = 0.187, *p* = 0.668). The analysis revealed that there was no statistically significant interaction between the time point and group allocation (F(2, 31) = 0.952, *p* = 0.397).

### 3.3. Influence of User Engagement on the Treatment Effect

A multiple linear regression analysis was calculated to predict THI change based on the user interaction (i.e., how often did the participant rate a tip) and total app usage (i.e., how often did the participant fill out a questionnaire). The interaction of the factors ‘user interaction‘ and‚ total usage’ was a significant predictor of THI change (Table 3). The total usage was measured by the number of EMA questionnaires filled out by the participant. Within the app, the user had the possibility to rate the tips with one to five stars; this user interaction was measured by the number of tip ratings done by the participant. 

### 3.4. Analysis of Changes in Patient Empowerment

Health literacy. A two-way mixed model ANOVA was performed to analyze the effect of the time point (pre vs. post) and group allocation on the health literacy subscore. Analysis of the main effects revealed that the group allocation did not have a statistically significant effect on the health literacy subscore (F(2, 33) = 0.455, *p* = 0.639) while the influence of the time point was statistically significant (F(1, 30) = 10.1, *p* = 0.004). The analysis revealed that there was no statistically significant interaction between the time point and group allocation (F(2, 30) = 0.350, *p* = 0.708).

For post hoc analysis, a Wilcoxon Signed Rank Test was calculated and revealed a statistical significance between the pre- and post-intervention phase (W = 428, *p* = 0.042). Cohen’s d effect size was calculated with 0.566 (confidence interval: 0.076 to 1.06). 

Coping. A two-way mixed model ANOVA was performed to analyze the effect of the time point (pre vs. post) and group allocation on the coping subscore. Analysis of the main effects revealed that the group allocation did not have a statistically significant effect on the coping subscore (F(2, 33) = 0.310, *p* = 0.736) while the influence of the time point was statistically significant (F(1, 30) = 4.59, *p* = 0.040). The analysis revealed that there was no statistically significant interaction between the time point and group allocation (F(2, 30) = 0.817, *p* = 0.451).

For post hoc analysis, a Wilcoxon signed rank test was calculated and revealed statistical significance between the pre- and post-intervention phase (W = 492, *p* = 0.216). Cohen’s d effect size was calculated with 0.285 (confidence interval: −0.198 to 0.769). 

Confidence, Self efficacy, and Self awareness. Two-way mixed model ANOVA was performed to analyze the effect of the time point (pre vs. post) and group allocation on the subscores of confidence, self efficacy, and self awareness. Analysis of main effects and interaction effects did not reveal any statistically significant influence of time point or group allocation on these subscores. The results are reported in Table 4.

## 4. Discussion

The main objective of this feasibility study was to test a novel smartphone-guided educational treatment with self-help tips for people with chronic tinnitus who are using the app unsupervised during their daily routine. Participants used the smartphone app for a period of four months in their everyday life without guidance from or personal contact with the study personal. Therefore, the study design represents a naturalistic setting of users that download a treatment app from the App Store and use the health app without medical guidance.

Changes in tinnitus. Comparing the tinnitus outcome measures before and after smartphone-guided treatment, we measured changes in the tinnitus-related distress, but not in tinnitus loudness. Improvements on the Tinnitus Severity numeric rating scale reached an effect size of 0.408, while the improvements on the THI were much smaller with an effect size of 0.168. The changes in the tinnitus loudness have been far from any meaningful change. This suggests that the TinnitusTipps treatment helped to reduce psychological suffering while the tinnitus sound itself is still present and remains unchanged. 

Importantly, the main effect for group did not reach the significance level in either analysis, suggesting no difference between a treatment duration of two months versus four months. Future studies will be needed to identify the best treatment duration for the intervention. It is possible that the duration of two months is already enough to reach the maximum possible treatment effect. 

In this study, the tinnitus assessment was made at the start and the end of the app-based educational counseling. No follow-up assessment was made in this feasibility study. We therefore observed only short-term effects. For future studies, we also recommend including follow-up assessments after six and 12 months. 

Relationship between user behavior and tinnitus change. The individual results of the participants had great variability. While some participants reported a strong reduction of tinnitus suffering during the treatment period, other participants reported no change or even worsening. An additional analysis of user behavior revealed that the amount of app usage (i.e., count how often the participant filled out the questionnaires) and the user interaction (i.e., how often did the user rate the tips) can predict tinnitus improvement. Interestingly, neither the main effect of total usage nor the main effect of user interaction explained the variance of the tinnitus changes significantly. The interaction effect of total usage and interaction, however, explained the changes in tinnitus distress significantly. This means that the frequent and intensive use of the app is a crucial factor for treatment success: participants that used the app more often and interacted with the app intensively, reported a stronger improvement of the tinnitus. We suggest that future studies track more details of the user behavior to allow additional analyses on the relationship between user behavior and tinnitus improvement. If intensive use of the health app was linked to better tinnitus improvement, one might hypothesize about a causal interaction and consequently try to increase the attractiveness of the app and motivate users to make more regular use of the app. After end of treatment, participants were asked open-ended questions about how to improve the app. Individual users reported that they would prefer fewer EMA questions during the day and that some questions (e.g., questions about hearing aids) did not fit their individual situation. This user feedback suggests that more individualized EMA sampling and fewer questionnaires could help to increase the app usage. This is important feedback when thinking about how to increase the total app usage. Sending the user more prompts might not help to increase the app usage. It might even have the opposite effect when the users become annoyed by too many prompts per day. In this context, it also needs to be considered that the prompts also increase the awareness toward the tinnitus for a short amount of time. In an analysis from 2016 on the TrackYourTinnitus app [22], it was shown on a larger group of users that repeated asking about tinnitus did not change the tinnitus distress in the long run. However, this does not exclude the possibility that the repeated measurements can have an effect, in either direction, for an individual person. Further research is needed here and the scientific work would benefit from a method that can measure tinnitus without asking for it, i.e., without raising awareness for tinnitus. 

Changes in patient empowerment. Patient empowerment was measured with the custom questionnaire “Tinnitus Empowerment Scale (TES)”. The subscales ‘health literacy’ and ‘coping’ revealed a significant improvement following the smartphone-guided educational treatment with effect sizes of 0.566 and 0.285, respectively. No significant improvements were found for the subscales “confidence”, “self efficacy”, and “self awareness”. The TES is a new scale that was developed and tested in this study. Validation data and norm tables do not exist yet. Therefore, the results need to be interpreted with caution and more research is needed for the development of the tool. With this caution in mind, it can be summarized that there are hints for improvement of self-rated health literacy and tinnitus-related coping strategies following the use of smartphone-guided educational treatment. For future studies, we would also suggest measuring empowerment at several time points during the treatment. This could be used to describe the health journey in more detail and investigate whether certain aspects of empowerment are more important at specific time points of the health journey (see e.g., [29]). As an example, with a larger sample, it would be possible to analyze whether tinnitus duration, the number of comorbidities, or traumatic life events have an influence. 

Dropouts. Between study allocation and final assessment, 26 of 62 participants dropped out of the study. The reasons are listed in Figure 1. A closer analysis of the reasons showed that in 16 of the 26 participants, the reasons for dropout could be addressed by technical improvements to the app. A leading cause for dropout was the installation process of the app via testflight, which was more complicated than the installation of an ordinary app via the appstore. Furthermore, participants dropped out because the app did not run on their smartphone (e.g., older iOS version or Android). Although technical requirements were clearly communicated during the recruitment process, several participants ended up in the study without a suitable smartphone. It needs to be mentioned here that the technical issues might also lead to a selection bias of the study. Only those participants that were able to overcome the technical barriers remained in the study. In general, such a bias reduces the generalizability of the study. Therefore, future studies need to address these technical issues if they provide an app for both systems, iOS and Android, and make the app available via the respective appstores. 

### Conclusion and Future Directions

With this study, we explored the feasibility of providing educational training on the smartphone without the guidance of a medical person. We observed small to medium improvements in tinnitus distress measures, but not on tinnitus loudness. Participants with a greater commitment to the intervention showed stronger improvements in tinnitus distress. It can be hypothesized that these improvements are related to the observed improvements in tinnitus-specific health literacy and coping strategies. Technical and methodological improvements for future work in this area is discussed. With the use of modern smartphone technology, there is a huge potential that can be unlocked in future studies: e.g., with more detailed knowledge of the individual patient and knowledge of large-scale crowdsensing patient data, it would be possible to deliver personalized tips and counseling to the participant [30]. Furthermore, with a continuous evaluation of symptoms related to tinnitus, it would be possible to predict the health condition of the individual person with tinnitus [23] and deliver tips and tinnitus-specific knowledge to the participant at times when it is most important. 

One restriction of these innovative eHealth solutions via smartphone apps remains: These solutions are always limited to people that own a smartphone and are willing and able to use it for treatment. To a certain extent, this always introduces a selection bias to these kinds of studies. 

## Figures and Tables

**Figure 1 jcm-11-01825-f001:**
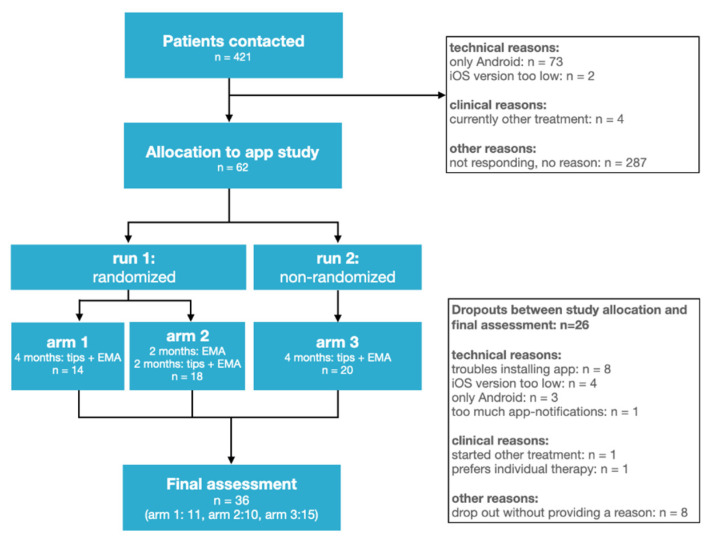
Participant flow diagram. In treatment arm 1 and 3, participants received four months of smartphone-guided educational counseling treatment plus Ecological Momentary Assessment. In treatment arm 2, participants received for two months, Ecological Momentary Assessment only, followed by a period of another two months where the participants received the smartphone-guided educational counseling treatment plus Ecological Momentary Assessment.

**Table 1 jcm-11-01825-t001:** Characteristics of participants in final analysis. *p* values is based on Chi square test for count variables or Kruskal–Wallis test for continuous variables. The sum scores of the THI range from 0 to 100. The scores for NRS the tinnitus distress range from 1 to 5, while the scores for the NRS tinnitus loudness range from 0 to 10.

Characteristics	Units	Full Cohort	Arm 1	Arm 2	Arm 3	*p*-Value
Participants per protocol	# Participants	36	11	10	16 15	0.433
Gender: female	# Participants (%)	14 (38.9)	4 (36.3)	4 (40.0)	6 (40.0)	0.752
Gender: male	# Participants (%)	22 (61.1)	7 (63.6)	6 (60.0)	9 (60.0)	0.728
Tinnitus duration	Years [mean (SD)]	8.6 (17.5)	14.1 (28.6)	9.8 (10.0)	3.2 (5.4)	0.054
Age at baseline	Years [mean (SD)]	49.4 (11.7)	49.6 (9.8)	51.7 (12.7)	47.6 (12.6)	0.381
THI at baseline	Points [mean (SD)]	48.1 (23.6)	50.7 (22.2)	44.4 (24.7)	48.6 (25.4)	0.826
NRS tinnitus distress at baseline	Points [mean (SD)]	2.5 (1.0)	2.5 (1.0)	2.8 (0.9)	2.2 (1.0)	0.400
NRS tinnitus loudness at baseline	Points [mean (SD)]	6.3 (2.2)	6.0 (2.1)	7.0 (2.6)	6.2 (2.1)	0.349

**Table 2 jcm-11-01825-t002:** Results of the mixed effects models for tinnitus outcome measures.

Outcome	Fixed Effects				Random Effects	
		numDF/denDF	*F*-Value	*p*-Value		SD
THI sum score	Time	1/31	3.94	0.056	participants	22.1
	Group	2/33	0.270	0.765	residuals	7.41
	Time * Group	2/31	0.403	0.672		
TS1: tinnitus problem	Time	1/31	14.0	<.001	participants	0.858
	Group	2/33	0.862	0.432	residuals	0.397
	Time * Group	2/31	3.14	0.057		
TS2: tinnitus loudness	Time	1/31	0.187	0.668	participants	1.85
	Group	2/33	0.265	0.769	residuals	1.25
	Time * Group	2/31	0.952	0.397		

**Table 3 jcm-11-01825-t003:** User engagement of enrolled participants.

Outcome					
		Estimate	SE	*t*-Value	*p*-Value
THI sum score	User interaction	−1.41	2.07	−0.682	0.501
	Total usage	1.87	2.03	0.918	0.366
	User interaction * Total usage	−4.63	1.57	−2.96	0.006

**Table 4 jcm-11-01825-t004:** Results of the mixed effects models for patient empowerment outcome measures.

Subscores	Fixed Effects				Random Effects	
		numDF/denDF	*F*-Value	*p*-Value		SD
Health Literacy	Time	1/30	10.1	0.004	participants	1.35
	Group	2/33	0.455	0.639	residuals	1.39
	Time * Group	2/30	0.350	0.708		
Coping	Time	1/30	4.59	0.040	participants	2.47
	Group	2/33	0.310	0.736	residuals	1.22
	Time * Group	2/30	0.817	0.451		
Confidence	Time	1/30	0.041	0.840	participants	1.49
	Group	2/33	1.02	0.373	residuals	1.30
	Time * Group	2/30	0.667	0.521		
Self Efficacy	Time	1/30	1.62	0.213	participants	2.55
	Group	2/33	1.01	0.375	residuals	1.32
	Time * Group	2/30	0.416	0.663		
Self Awareness	Time	1/30	2.95	0.096	participants	1.83
	Group	2/33	2.26	0.120	residuals	1.67
	Time * Group	2/30	0.744	0.484		

## Data Availability

The data presented in this study are available on reasonable request from the corresponding author.

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
