# Peer review of "Smartphone-Guided Educational Counseling and Self-Help for Chronic Tinnitus"

_jcm, 2022, doi:10.3390/jcm11071825_

Round 1
Reviewer 1 Report
1. By focusing on tinnitus, the authors are addressing an important issue that remains relatively poorly understood but which can create extreme distress for a given individual. Overall, the paper is well written with some grammatical issues that appear to be related to translation into English. The background and support for the project is also quite well developed. There are, however, some issues raised that limit the ease with which the intervention and the findings can be understood and interpreted.
2. Note is made that the “tips were designed to be concise and succinct so that they would preferably not exceed the display size of an average smartphone and that reading the tips would be possible without spending a lot of time in everyday life.”
This is reasonable although hard to picture and an average of 100 words does not appear to be too “concise”. An example, maybe even in a supplement, of the structure of one tip might be valuable.
3. Given the above note on conciseness, it was more difficult to get a clear picture of the Ecological Momentary Assessment. This approach, capturing an assessment as it happens, is positive and utilized in many health apps. However, it’s unclear exactly what was asked and how many questions.
The description notes that “Ecological Momentary Assessment (EMA) of the tinnitus was done using two questions on tinnitus loudness and tinnitus distress, using the questionnaire published by Schlee et al. 2016 [23]. In addition, patients were asked about their momentary hearing ability, their limitation due to hearing loss, and if they were using a hearing aid at the moment. Furthermore, they have been asked about the momentary perceived stress level, and exhaustion. The goal was to fill out the EMA questionnaire at three times during the day and the participants received random notifications to fill out the questions. The EMA data are reported in [24].
This appears to include a fairly lengthy list of items all done on the iphone 3x per day at random times. Are there data on how long it took to respond to these various screens? Did anything happen if an individual did not respond when sent to notice to fill these out and were these “missing data”?
4. In addition, the description of the tools used, which appear to only be asked at the start/baseline and end of the study, do not include the response options and range of scores with any cutoffs normally used to indicate levels of distress. This makes interpretation of Table 1 and the analytic findings difficult. Further description of the tools would be helpful. It does appear, after checking on the Tinnitus Handicap Index, the participants may have scored in the moderate area, yet the severity appears mild – depending on the scale used. It would also help to have some idea of the items used for “coping” since coping strategies can vary so broadly.
5. Given the relatively small “n”, was there a cap on the number of missing items before a given scale, or participant, was fully removed from the analysis?
6. In the discussion note is made that individuals used the app in “everyday life without guidance or reminders by the study personal” yet they received random requests to fill in the questionnaires 3 times a day which, just by virtual of that reminder, would alert them to the use of the app. Was this assessed?
7. Line 322-323: I don’t believe this can be noted as change in coping was not assessed.
“This suggests that TinnitusTipps treatment helped participants improve 322 their coping skills and reduce psychological suffering while the tinnitus sound itself is still 323 present and remains unchanged”
8. Line 325: “Neither” should be “either” - Importantly, the main effect for group did not reach the significance level in neither analysis..”
9. As a suggestion, I recommend using “persons with chronic tinnitus” rather than “chronic tinnitus patients”. E.g see line 226: “In total, 36 chronic tinnitus patients…” Under eligibility it is listed as “only patients with chronic tinnitus for 6 months….”
Author Response
Reviewer 1
1. By focusing on tinnitus, the authors are addressing an important issue that remains relatively poorly understood but which can create extreme distress for a given individual. Overall, the paper is well written with some grammatical issues that appear to be related to translation into English. The background and support for the project is also quite well developed. There are, however, some issues raised that limit the ease with which the intervention and the findings can be understood and interpreted.
>> We want to thank the reviewer for the very good and constructive feedback. This helped a lot to improve the manuscript. We addressed the concerns and provided a point-by-point response below.
2. Note is made that the “tips were designed to be concise and succinct so that they would preferably not exceed the display size of an average smartphone and that reading the tips would be possible without spending a lot of time in everyday life.”
This is reasonable although hard to picture and an average of 100 words does not appear to be too “concise”. An example, maybe even in a supplement, of the structure of one tip might be valuable.
>> Good idea. Merci. We added a screenshot of two examples to the supplemental material (S1) and provided the English translation for it.
3. Given the above note on conciseness, it was more difficult to get a clear picture of the Ecological Momentary Assessment. This approach, capturing an assessment as it happens, is positive and utilized in many health apps. However, it’s unclear exactly what was asked and how many questions.
The description notes that “Ecological Momentary Assessment (EMA) of the tinnitus was done using two questions on tinnitus loudness and tinnitus distress, using the questionnaire published by Schlee et al. 2016 [23]. In addition, patients were asked about their momentary hearing ability, their limitation due to hearing loss, and if they were using a hearing aid at the moment. Furthermore, they have been asked about the momentary perceived stress level, and exhaustion. The goal was to fill out the EMA questionnaire at three times during the day and the participants received random notifications to fill out the questions. The EMA data are reported in [24].
This appears to include a fairly lengthy list of items all done on the iphone 3x per day at random times. Are there data on how long it took to respond to these various screens? Did anything happen if an individual did not respond when sent to notice to fill these out and were these “missing data”?
>> Thanks for this concern. In fact, the questionnaire was short and was designed in a way that it can be answered fast. Internal tests showed that the EMA questionnaire can be answered in less than a minute. We improved this in the current version of the manuscript. A screenshot of the EMA questionnaire, incl. the English translation, is added to the supplemental material S2.
4. In addition, the description of the tools used, which appear to only be asked at the start/baseline and end of the study, do not include the response options and range of scores with any cutoffs normally used to indicate levels of distress. This makes interpretation of Table 1 and the analytic findings difficult. Further description of the tools would be helpful. It does appear, after checking on the Tinnitus Handicap Index, the participants may have scored in the moderate area, yet the severity appears mild – depending on the scale used. It would also help to have some idea of the items used for “coping” since coping strategies can vary so broadly.
>> Thanks a lot for making us aware of this missing information! We added the information on the range of scores to the legend of table 1 ("The sum scores of the THI range from 0 to 100. The scores for NRS the tinnitus distress range from 1 to 5, while the scores for the NRS tinnitus
loudness range from 0 to 10."). Also, we added the Tinnitus Empowerment Questionnaire, including the questions on coping, to the supplemental material S3.
5. Given the relatively small “n”, was there a cap on the number of missing items before a given scale, or participant, was fully removed from the analysis?
>> In fact, the number of missing items was very low. This was due to the fact, that the participants got reminded electronically whenever they missed to answer an item.
6. In the discussion note is made that individuals used the app in “everyday life without guidance or reminders by the study personal” yet they received random requests to fill in the questionnaires 3 times a day which, just by virtual of that reminder, would alert them to the use of the app. Was this assessed?
>> Good point. This was indeed not clearly written. Digital reminders are also a form of reminder. We rephrased this part of the sentence accordingly "... without guidance from or personal contact with the study personal".
7. Line 322-323: I don’t believe this can be noted as change in coping was not assessed.
“This suggests that TinnitusTipps treatment helped participants improve 322 their coping skills and reduce psychological suffering while the tinnitus sound itself is still 323 present and remains unchanged”
>> We agree with this criticism and removed the part about the coping skills. Thanks for this correction.
8. Line 325: “Neither” should be “either” - Importantly, the main effect for group did not reach the significance level in neither analysis..”
>> Thanks for spotting this error! It is corrected.
9. As a suggestion, I recommend using “persons with chronic tinnitus” rather than “chronic tinnitus patients”. E.g see line 226: “In total, 36 chronic tinnitus patients...” Under eligibility it is listed as “only patients with chronic tinnitus for 6 months....”
>> Thanks a lot for this comment. We changed this wording in the manuscript. Also, we replaced "patients" by "participants" or "users", when appropriate.

Reviewer 2 Report
Comments to Author(s)
“Smartphone-guided educational counseling and self-help for chronic tinnitus”
Summary
This study tests the feasibility of a smartphone-based app to provide educational counseling related to tinnitus. The “TinnitusTipps” app includes daily smartphone-guided tips on tinnitus management as well as Ecological Momentary Assessments that ask the user to respond to various items around tinnitus loudness/distress, general stress level, and exhaustion. Results suggest that the app helped with tinnitus-related distress, but not with tinnitus loudness. I hope the author(s) find the following comments useful in revisions.
Comments
Abstract
- If space allows, it would help the reader to provide more detailed information in this section to better understand the study.
Introduction
- Describe the successful treatments for tinnitus patients (e.g., CBT)
- Is “educative counseling” the most appropriate label for the strategy outlined in the paper? Perhaps “educational counseling”?
- Elaborate on the three challenges to studies of educational counseling because these seem like limitations for the current study.
- Has educational counseling been useful in other health care settings (e.g., for hearing loss or vestibular outcomes)?
Materials and Methods
- Overall, the organization of this section could be improved to help the reader better understand the study design. As I read through the section, I found that information I needed to, for example, understand the study design was not presented until later in the section on participants. Consider rearranging some of the paragraphs to describe the relevant information in the same section (e.g., call the section Study Design and Participants, or some such).
Study Design
- Minor: check throughout for spelling of “counseling.”
- Describe the recruitment – were patients recruited from audiology, otolaryngology, general clinic, etc.? Were they there for a tinnitus appointment, or any type of ear-related appointment? Any inclusion/exclusion criteria (e.g., age, hearing ability, level of tinnitus, language, etc.)? How were participants put into run 1 versus run 2? Some of this information is brought up in a later paragraph (Participants), but perhaps the two paragraphs should be combined.
- Figure: To help the reader, consider adding a few anchor words to the text already in the boxes. For example, instead of “run 1” and “run 2,” perhaps write “randomized group” and “nonrandomized group.” Similarly, how might the “arm” boxes be described? Perhaps, Arm 1: 4 months app + EMA, Arm 2: EMA, then EMA + app, Arm 3: app + EMA.
- The last sentence of this paragraph (“The aim was to test the feasibility of app-based 110 educational counseling treatment on a clinical tinnitus population where the participants use the smartphone unsupervised during their daily routine”) is important and might be more impactful in the current study paragraph (last paragraph of background section).
Smartphone-guided educational counseling
- If space allows, it would neat to see a screenshot of one of the tips as a figure! This would help the reader better understand the “title” and “objective” points described in the text as well as see what the user experienced every day.
- Minor point: typo of “tipp.”
Ecological Momentary Assessment
- Please elaborate on perceived stress level and exhaustion – how were these items measured? It also might help to write out the items on tinnitus loudness and distress. I think you describe this later in the Measurement instruments section – is it necessary to make this a separate section, or does it belong in the EMA section?
- Perhaps create a supplementary figure showing what the EMA portion of the intervention looks like in the app?
Statistical analysis
- I imagine a lot of people did not complete the EMA when prompted, or that responses to EMA declined over time. It would be helpful to discuss missingness in more detail because this could influence the results.
Table 1
- I am intrigued by the differences in tinnitus duration across study arms. Was this intentional, or did the randomization simply end up assigning people with longer durations into Arm 1 and shorter durations into Arm 3? This could be an important factor to consider because time spent living with tinnitus could impact how individuals respond to treatment of any kind.
Analysis of changes in patient empowerment
- Please define health literacy, coping, and confidence/self-efficacy/self-awareness in the methods section before introducing these concepts in the results. It would also help to justify in the background section why these measures were analyzed.
Discussion
- Changes of tinnitus: it is not clear to me that the app improved coping skills and reduced psychological suffering. Please expand on this connection.
- Changes of tinnitus: I appreciate that there was no difference between two versus four months of study interaction. However, might the study results simply be showing a short-term rather than long-term effect? How might participants rate their tinnitus after six months, 12 months? The results might be reflecting individuals’ willingness to engage in a novel app for a short period of time, but not show how they will behave for a longer period of time.
- Dropouts: Losing around 42% of the sample is pretty significant. Technical issues are naturally a barrier, but the people who can overcome technical issues are more selective than those who drop out due to technical issues. How do we know whether this amount of attrition is not driving the study results?
- Additional discussion of the limitations of this study is warranted. Aside from those limitations I have discussed in various points above, who has access to smartphone technology and who does not?
Author Response
Reviewer 2
This study tests the feasibility of a smartphone-based app to provide educational counseling related to tinnitus. The “TinnitusTipps” app includes daily smartphone-guided tips on tinnitus management as well as Ecological Momentary Assessments that ask the user to respond to various items around tinnitus loudness/distress, general stress level, and exhaustion. Results suggest that the app helped with tinnitus-related distress, but not with tinnitus loudness. I hope the author(s) find the following comments useful in revisions.
>> We want to thank the reviewer for the detailed and thoughtful comments, which were indeed very useful and helped to improve the manuscript tremendously. We provide a point-by-point response below.
Comments
Abstract
• If space allows, it would help the reader to provide more detailed information in this section to better understand the study.
>> We agree. Unfortunately, we already reached the maximum of 200 words that are allowed for the abstract in the JCM.
Introduction
• Describe the successful treatments for tinnitus patients (e.g., CBT)
>> Thanks you for this idea. We expanded this part of the introduction and described CBT in more detail.
• Is “educative counseling” the most appropriate label for the strategy outlined in the paper? Perhaps “educational counseling”?
>> Thank you for this nice suggestion. We changed it accordingly. It is much better this way.
• Elaborate on the three challenges to studies of educational counseling because these seem like limitations for the current study.
>> Thank you for this feedback. In the revised version of the manuscript, we elaborate much more on these three challenges (lines 89 - 114).
• Has educational counseling been useful in other health care settings (e.g., for hearing loss or vestibular outcomes)?
>> This is an interesting question. We did a literature search on it and did not find any studies reporting effects of educational counseling in other hearing-related conditions. In some study designs, educational counseling is used as active control condition. Maybe there is a new research area to be conquered ;).
Materials and Methods
• Overall, the organization of this section could be improved to help the reader better understand the study design. As I read through the section, I found that information I needed to, for example, understand the study design was not presented until later in the section on participants. Consider rearranging some of the paragraphs to describe the relevant information in the same section (e.g., call the section Study Design and Participants, or some such).
>> Indeed, the order was a bit misleading. We like your suggestion, rearranged the order and called the first section "Study design and participants".
Study Design
• Minor: check throughout for spelling of “counseling.”
>> Thanks. We corrected it throughout the manuscript.
• Describe the recruitment – were patients recruited from audiology, otolaryngology, general clinic, etc.? Were they there for a tinnitus appointment, or any type of ear-related appointment? Any inclusion/exclusion criteria (e.g., age, hearing ability, level of tinnitus, language, etc.)? How were participants put into run 1 versus run 2? Some of this information is brought up in a later paragraph (Participants), but perhaps the two paragraphs should be combined.
>> Indeed, the order was a bit strange. We rearranged the order and called the first section "Study design and participants".
• Figure: To help the reader, consider adding a few anchor words to the text already in the boxes. For example, instead of “run 1” and “run 2,” perhaps write “randomized group” and “nonrandomized group.” Similarly, how might the “arm” boxes be described? Perhaps, Arm 1: 4 months app + EMA, Arm 2: EMA, then EMA + app, Arm 3: app + EMA.
>> Thank you for the nice suggestion. We used the ideas and improved the figure.
• The last sentence of this paragraph (“The aim was to test the feasibility of app-based 110 educational counseling treatment on a clinical tinnitus population where the participants use the smartphone unsupervised during their daily routine”) is important and might be more impactful in the current study paragraph (last paragraph of background section).
>> Thank you for this idea. We followed your suggestion.
Smartphone-guided educational counseling
• If space allows, it would neat to see a screenshot of one of the tips as a figure! This would help the reader better understand the “title” and “objective” points described in the text as well as see what the user experienced every day.
>> Nice idea! We added two example screenshots in the supplemental material (S1).
• Minor point: typo of “tipp.”
>> Thanks a lot. We corrected this typo. Ecological Momentary Assessment
• Please elaborate on perceived stress level and exhaustion – how were these items measured? It also might help to write out the items on tinnitus loudness and distress. I think you describe this later in the Measurement instruments section – is it necessary to make this a separate section, or does it belong in the EMA section?
>> We added more detailed information on these measurements in the supplemental material section. The original German version and an English translation is provided.
• Perhaps create a supplementary figure showing what the EMA portion of the intervention looks like in the app?
>> Wonderful idea. Thanks. We added this to the supplemental material (S2).
Statistical analysis
• I imagine a lot of people did not complete the EMA when prompted, or that responses to EMA declined over time. It would be helpful to discuss missingness in more detail because this could influence the results.
>> Yes, this can indeed influence the results. If you turn it around: those that missed more prompts, used the app less often. In part 3.3 of the manuscript, we showed that the the total app usage and the user interaction influence the results of the THI sum score (interaction effect). The topic of the prompts is interesting: Prompts are sent to remind the user and increase app usage. At the same time, the participants gave us the feedback that there were already too many prompts. Therefore, an increase of prompts might not help to increase the app usage - it might even have an opposite effect. We added this point to the discussion section. Thanks a lot for bringing this up.
Table 1
• I am intrigued by the differences in tinnitus duration across study arms. Was this intentional, or did the randomization simply end up assigning people with longer durations into Arm 1 and shorter durations into Arm 3? This could be an important factor to consider because time spent living with tinnitus could impact how individuals respond to treatment of any kind.
>> This was also surprising for us. We do not know the reason for the shorter tinnitus duration in arm 3. It is most likely a random recruitment effect.
Analysis of changes in patient empowerment
• Please define health literacy, coping, and confidence/self-efficacy/self-awareness in the methods section before introducing these concepts in the results. It would also help to justify in the background section why these measures were analyzed.
>> We agree that this was not clear in the first version of the manuscript. Especially because the details of the questionnaire were not reported. We now added the full questionnaire, to the supplemental material S3.
Discussion
• Changes of tinnitus: it is not clear to me that the app improved coping skills and reduced psychological suffering. Please expand on this connection.
>> Thanks for mentioning this. We corrected the discussion. This was an interpretation that is not supported by the data. We took it out.
• Changes of tinnitus: I appreciate that there was no difference between two versus four months of study interaction. However, might the study results simply be showing a short- term rather than long-term effect? How might participants rate their tinnitus after six months, 12 months? The results might be reflecting individuals’ willingness to engage in a novel app for a short period of time, but not show how they will behave for a longer period of time.
>> This is a very important point!!! Thanks. We added this to the discussion section and suggest that studies on this topic should include follow-up assessments.
• Dropouts: Losing around 42% of the sample is pretty significant. Technical issues are naturally a barrier, but the people who can overcome technical issues are more selective than those who drop out due to technical issues. How do we know whether this amount of attrition is not driving the study results?
>> We think that you addressed a very important point. Thank you. We added this point to the discussion section. "It needs to be mentioned here that the technical issues might also lead to a selection bias of the study. Only those participants that were able to overcome the technical barriers remained in the study. In general, such a bias reduces the generalizability of the study. Therefore, future studies need to address these technical issues if they provide an app for both systems, iOS and Android, and make the app available via the respective appstores."
• Additional discussion of the limitations of this study is warranted. Aside from those limitations I have discussed in various points above, who has access to smartphone technology and who does not?
>> This is an important and good point. We added this to the discussion at the very end (lines 504-507). Another limitation is added here as well: Some persons might own a smartphone but are not willing or not able to use it for treatment.

Round 2
Reviewer 1 Report
I appreciate the authors careful review of the reviewer comments and the changes that were incorporated to address the feedback. The changes enhance the clarity of the paper and its findings. There are just a few minor issues that would benefit from possible editing.
First, I noted that, in the abstract, the authors use the acronym “THI” without prior use or writing it as the Tinnitus Handicap Index. If feasible, it would be helpful to write this out the first time here and in the text as necessary (“Improvements on the Tinnitus Severity numeric rating scale reached an effect size of .408, while the improvements on the THI were much smaller with an effect size of .168.”)
Lines 88-114: There is a discussion of what challenges that to be considered (There are three major challenges with studies on educational counseling. The first challenge is the choice of a control condition)
This is a helpful discussion but appears out of context – the design of the study is not yet discussed and introduces the app by name before the idea of an app is presented and discussed. I suggest moving this whole section to immediately after the paragraph explaining what the study is doing along with the aim which could include the name of the app so its mention subsequently is clear. E.g. put the paragraph right before “materials and methods”. The “name” of the app can be given (which seems to be “TinnitusTipps”) and the explanation of the choices made during implementation. It could also be incorporated within the section on methods.
Discussion: I realized in re-reading this section that the intent of this sentence isn’t clear to me:
Interestingly, neither the main effect of total usage nor the main effect of user interaction explained the variance of the tinnitus changes significantly.
As a question, one of the findings is that fairly intense engagement with or use of the app supported better outcomes but the range of impact varied with some reporting a worsening. Do the data support at all that the engagement itself, by raising my awareness of the presence of tinnitus, actually make it seem worse? I now think about it more and it’s on my mind more? Maybe needs further study. I believe this is alluded to at the end but is a valuable issue to explore in order to target any intervention appropriately.
Author Response
Response to Reviewer 1, Round 2.
R: I appreciate the authors careful review of the reviewer comments and the changes that were incorporated to address the feedback. The changes enhance the clarity of the paper and its findings. There are just a few minor issues that would benefit from possible editing.
>> Once again, we want the thank the reviewer for the great suggestions and ideas for improvement. A point-by-point response is given below and the improved version of the manuscript uploaded.
First, I noted that, in the abstract, the authors use the acronym “THI” without prior use or writing it as the Tinnitus Handicap Index. If feasible, it would be helpful to write this out the first time here and in the text as necessary (“Improvements on the Tinnitus Severity numeric rating scale reached an effect size of .408, while the improvements on the THI were much smaller with an effect size of .168.”)
>> Thanks for the good suggestion! We changed it accordingly.
Lines 88-114: There is a discussion of what challenges that to be considered (There are three major challenges with studies on educational counseling. The first challenge is the choice of a control condition)
This is a helpful discussion but appears out of context – the design of the study is not yet discussed and introduces the app by name before the idea of an app is presented and discussed. I suggest moving this whole section to immediately after the paragraph explaining what the study is doing along with the aim which could include the name of the app so its mention subsequently is clear. E.g. put the paragraph right before “materials and methods”. The “name” of the app can be given (which seems to be “TinnitusTipps”) and the explanation of the choices made during implementation. It could also be incorporated within the section on methods.
>> Nice idea, thanks. It is changed now. Also we added the name of the TinnitusTipps app in this section.
Discussion: I realized in re-reading this section that the intent of this sentence isn’t clear to me:
Interestingly, neither the main effect of total usage nor the main effect of user interaction explained the variance of the tinnitus changes significantly.
As a question, one of the findings is that fairly intense engagement with or use of the app supported better outcomes but the range of impact varied with some reporting a worsening. Do the data support at all that the engagement itself, by raising my awareness of the presence of tinnitus, actually make it seem worse? I now think about it more and it’s on my mind more? Maybe needs further study. I believe this is alluded to at the end but is a valuable issue to explore in order to target any intervention appropriately.
>> Thanks for mentioning this. We included this topic in the discussion as well. In an analysis from 2016 on the TrackYourTinnitus app[23], it was shown on a larger group of users, that the repeated asking about tinnitus did not change the tinnitus distress in the long run. However, this doesn't exclude the possibility that the repeated measurements can have an effect, in either direction, for an individual person. Further research is needed here and the scientific work would benefit from a method that can measure tinnitus without asking for it, i.e. without raising awareness for tinnitus.
